# Association between Taxonomic Composition of Gut Microbiota and Host Single Nucleotide Polymorphisms in Crohn’s Disease Patients from Russia

**DOI:** 10.3390/ijms24097998

**Published:** 2023-04-28

**Authors:** Maria Markelova, Anastasia Senina, Dilyara Khusnutdinova, Maria Siniagina, Elena Kupriyanova, Gulnaz Shakirova, Alfiya Odintsova, Rustam Abdulkhakov, Irina Kolesnikova, Olga Shagaleeva, Svetlana Lyamina, Sayar Abdulkhakov, Natalia Zakharzhevskaya, Tatiana Grigoryeva

**Affiliations:** 1Institute of Fundamental Medicine and Biology, Kazan (Volga Region) Federal University, 420008 Kazan, Russia; mimarkelova@gmail.com (M.M.);; 2Municipal Polyclinic №21, 420139 Kazan, Russia; 3Republican Clinical Hospital, 420064 Kazan, Russia; 4Hospital Therapy Department, Kazan State Medical University, 420012 Kazan, Russia; 5Lopukhin Federal Research and Clinical Center of Physical-Chemical Medicine of Federal Medical Biological Agency, 119435 Moscow, Russia; 6Molecular Pathology of Digestion Laboratory, A.I. Yevdokimov Moscow State University of Medicine and Dentistry, 127473 Moscow, Russia

**Keywords:** Crohn’s disease, microbiota, SNP

## Abstract

Crohn’s disease (CD) is a chronic relapsing inflammatory bowel disease of unknown etiology. Genetic predisposition and dysbiotic gut microbiota are important factors in the pathogenesis of CD. In this study, we analyzed the taxonomic composition of the gut microbiota and genotypes of 24 single nucleotide polymorphisms (SNP) associated with the risk of CD. The studied cohorts included 96 CD patients and 24 healthy volunteers from Russia. Statistically significant differences were found in the allele frequencies for 8 SNPs and taxonomic composition of the gut microbiota in CD patients compared with controls. In addition, two types of gut microbiota communities were identified in CD patients. The main distinguishing driver of bacterial families for the first community type are *Bacteroidaceae* and unclassified members of the *Clostridiales* order, and the second type is characterized by increased abundance of *Streptococcaceae* and *Enterobacteriaceae*. Differences in the allele frequencies of the rs9858542 (*BSN*), rs3816769 (*STAT3*), and rs1793004 (*NELL1*) were also found between groups of CD patients with different types of microbiota communities. These findings confirm the complex multifactorial nature of CD.

## 1. Introduction

Crohn’s disease (CD) is a chronic relapsing disease characterized by inflammation of various regions of the gastrointestinal tract, mainly the small and large intestines. While the etiology of this disease is still unclear, it is known to be multifactorial. The pathogenesis of CD depends on environmental factors, genetic predisposition, individual immune response, and intestinal microbiota.

Through the development of DNA sequencing technology in recent decades, there is a significant amount of data on the intestinal microbiota in health and disease. It is known that inflammatory bowel diseases (IBD) mainly affect the regions of the gastrointestinal tract with the maximum density of the bacterial population (colon and small intestine). Many studies confirm the association of microbiota composition with IBD [1,2,3,4,5], including in the Russian population [6,7,8,9]. The microbiota of IBD patients is most often characterized by reduced alpha diversity, and a decrease in abundances of *Firmicutes* and *Bacteroidetes*, and an increase in *Proteobacteria* and *E. coli*, in particular. At the functional level, these changes lead to reduced levels of short-chain fatty acids (SCFAs), especially butyrate—anti-inflammatory metabolites produced by microbiota, and shifts in oxidative stress pathways and the secretion of toxins [10,11]. Many genetic polymorphisms associated with IBD are located in genes related to the host immune response and, in particular, interaction with the microbiota. The most studied is the mutation in the *NOD2* gene, which determines the immune response to the peptidoglycan of the bacterial cell wall [12]. Thus, patients with CD carrying a mutation in *NOD2* are characterized by an increased amount of adhesive bacteria and decrease in the representation of *Faecalibacterium* [13,14], and NOD2-deficient mice have an altered microbiome [15]. The *NOD2* gene product interacts with the *ATG16L1* gene product, mutations in which are also associated with CD [16]. TH1 and TH17 immune responses are increased in mice with the *ATG16L1* T300A variant, and the bacterial genera *Bacteroides* and *Escherichia* are more prevalent in the intestines of these mice [17,18]. Mutations in the *FUT2* and *CARD9* genes also affect the gut microbiota of patients with IBD [18,19,20].

It is known that CD is accompanied by disturbed integrity of the intestinal wall due to inflammation [21]. This leads to an increased intestinal permeability and penetration of food and microbial antigens into the bloodstream, which is characterized by elevated serum antibodies against them. The most frequently mentioned immunoglobulins are anti-Saccharomyces cerevisiae antibodies (ASCA), which are significantly increased in the serum of patients with CD, coeliac disease, rheumatoid arthritis, and autoimmune liver diseases [22,23,24]. ASCA is known to be elevated even in unaffected relatives of IBD patients, probably due to a genetic predisposition to abnormal intestinal permeability [25]. In addition, bacteria can affect barrier function by degradation of the mucus layer, regulation of epithelial cell apoptosis, and synthesis of components necessary for tight junctions [26]. Thus, microbiota antigens can interact with the host immune system not only on the intestinal mucosa, but also in the bloodstream due to the impaired barrier function.

However, identifying human genome and microbiome associations in IBD patients remains an urgent task for a deeper understanding of the disease pathogenesis and further personalized treatment selection.

## 2. Results

### 2.1. Human Subjects

The present study involved 96 patients with Crohn’s disease (53 female and 43 male, mean age 32.3 ± 11.8 years). CD was diagnosed using standard clinical, endoscopic, and histological criteria. All patients were in the acute stage with varying severity and localization of inflammation (Table 1). The average duration of the CD was 9.3 ± 4.5 years. The control group consisted of 24 healthy volunteers (15 female and 9 male, mean age 35.3 ± 10.0 years).

### 2.2. Microbiota Analysis

#### 2.2.1. Gut Microbiota of CD Patients and Healthy Volunteers

The number of sequencing read pairs obtained from fecal samples of CD patients and healthy controls ranged from 53,175 to 182,362 (median 91,061). Raw reads were deposited in the NCBI SRA under accession number PRJNA938107 in the fastq format. After merging, quality control, removing of chimeric reads, and rarefying, 20,938 reads per sample remained.

The major bacterial phyla constituting the intestinal microbiota of healthy volunteers and CD patients were *Firmicutes* (65.2 ± 14.7% and 63.0 ± 16.6%, respectively), *Bacteroidetes* (22.9 ± 15.5% and 19.8 ± 17.5%), *Proteobacteria* (2.5 ± 3.2% and 7.3 ± 10.9%), and *Actinobacteria* (6.8 ± 6.0% and 6.9 ± 8.5%). 

Shannon’s alpha diversity index and observed operational taxonomic units (OTUs) were significantly reduced in CD patients compared with controls (Figure 1A). A decreased abundance of the families *Clostidiaceae*, *Coriobacteriaceae*, and *Rikenellaceae* and an increased representation of *Lactobacillaceae*, *Enterococcaceae*, *Streptococcaceae*, and *Enterobacteriaceae* were also found (Figure 1B).

Depending on the CDAI, differences in the taxonomic composition of CD patients’ microbiota are revealed. The *Firmicutes* phylum and *Micrococcaceae* and *Enterococcaceae* families showed significant positive correlations with CD activity (Figure 2A). Significant negative correlations with the severity of the disease were found for the *Bacteroidetes* phylum and *Eryspelotrichaceae*, [*Odoribacteraceae*], *Rikenellaceae*, *Coriobacteriaceae*, *Bacteroidaceae*, and *Porphyromonadaceae* families (Figure 2A). When CD patients were divided into three groups according to the activity of the disease, significant differences in the representation of three families were revealed—*Micrococcaceae* increased with the increase in CD severity, while the abundance of *Coriobacteriaceae* and *Bacteroidaceae* decreased (Figure 2B).

Based on the Dirichlet multinomial mixtures method, two types of microbiota can be distinguished according to the taxonomic composition (Figure 3). The first community type (I) included 61 CD patients and all 24 controls, while the second group (II) included 35 CD patients. Thus, the frequency of occurrence of microbiota types in CD patients and controls is significantly different (*p* = 0.0003, Exact Fisher test). The main driver representatives (the most abundant in these communities) of the first community type are the families *Lachnospiraceae*, *Ruminococcaceae*, and *Bacteroidaceae*, and an unclassified member of the order *Clostridiales* (Figure 4A), while the second type is determined by *Lachnospiraceae*, *Streptococcaceae*, *Ruminococcaceae*, and *Enterobacteriaceae* (Figure 4B).

#### 2.2.2. Analysis of Microbiota Community Types in CD Patients

When comparing the two types of communities identified in CD patients, the second type showed a significant decrease in the number of observed OTUs and Shannon’s alpha diversity index, indicating a more prominent dysbiosis (Figure 5A, Appendix A). A decrease in the abundance of the *Bacteroidetes* phylum and an increase in the *Proteobacteria*, *Fusobacteria*, and *Verrucomicrobia* phyla were also observed. Moreover, the abundance of the *Bacteroidaceae*, *Prevotellaceae*, *Lachnospiraceae*, *Ruminococcaceae*, and *Erysipelotrichaceae* families and unclassified *Clostridiales* were significantly declined in the second type of community of CD patients (Figure 5B, Appendix A). These bacteria are members of the normal human microbiota and play a role in maintaining intestinal homeostasis. An increased amount of the *Verrucomicrobiaceae*, *Enterococcaceae*, *Streptococcaceae,* and *Enterobacteriaceae* families was also found (Figure 5B, Appendix A). Thus, the microbiome of CD patients with community type II is characterized by prominent dysbiosis, while the microbiome of patients with the first type is more similar to the healthy ones.

#### 2.2.3. Analysis of Clinical Parameters in CD Patients with Different Types of Microbial Communities

When comparing CD patients with different types of microbial communities, no significant differences were found in clinical characteristics—duration of disease, location of inflammation (ileitis, colitis, ileocolitis), disease activity (based on the Crohn’s disease activity index), phenotype of disease (inflammatory, stricturing, fistulizing), or stool frequency (Table 2).

### 2.3. SNP Analysis

SNP Analysis in CD Patients and Healthy Volunteers

All 24 genetic markers agreed to Hardy–Weinberg equilibrium proportions in the control population (*p* > 0.05). Allele frequencies of 8 genetic polymorphisms were significantly different between the CD groups and healthy subjects (Table 3). The alleles rs1004819A and rs11209026G of the *IL23R* gene, as well as rs2241880A (*ATG16L1*), rs4958847A (*IRGM*), rs1992662G (*PTGER4*), rs2274910C (*ITLN1*), rs6601764T, and rs7807258C were found to be more frequent in patients with CD.

### 2.4. Analysis of Association between Microbiota and SNPs Allele Frequency in CD Patients

#### 2.4.1. SNP Analysis in CD Patients According to the Type of Microbial Community

In the group of CD patients with the second type of gut microbiota community, the following allele frequencies: A in rs9858542 of the *BSN* gene, T in rs3816769 of the *STAT3* gene, and C in rs1793004 of the *NELL1* gene were significantly increased (Table 4). All of these alleles are associated with an increased risk of CD [27,28,29,30,31,32,33].

#### 2.4.2. Correlation between SNP and Taxonomic Composition of Gut Microbiota in CD Patients

Statistically significant negative correlations of rs9858542 (*BSN*) with the number of observed OTUs and the representation of the *Bacteroidetes* phylum were revealed using an additive model (Figure 6). Rs3816769 (*STAT3*) showed a negative correlation with the phylum *Bacteroidetes* and especially with the family *Bacteroidaceae*. For rs1793004 (*NELL1*) a negative correlation was found with the family *Ruminococcaceae* and a positive correlation with the family *Enterococcaceae*. Furthermore, significant negative correlations were found between abundance of *Bacteroidaceae* with rs2274910 (*ITLN1*), rs2522057 (*IRF1-AS1*), rs224136 (intergenic), rs6908425 (*CDKAL1*), and rs12037606 (intergenic) and a positive correlations with rs1992662 (*PTGER4*), rs1456893 (intergenic), and 13361189 (*IRGM*). *Enterococcaceae* and *Enterobacteriaceae* families showed significant positive correlations with rs224136 (intergenic).

## 3. Discussion

Changes in the gut microbiota composition and role of genetics in CD patients have been described in a number of studies. However, there is limited data on CD patients in the Russian population. CD prevalence in Russia is estimated to be 3.0–7.88 cases per 100,000 population [34,35], and it rises 8–10% annually [35], but it is still substantially lower than in Western Europe and North America [36]. Patients in our study were recruited from two regions of Russia (the Republic of Tatarstan and Moscow), ensuring that people of different nationalities (mainly Russians and Tatars) were represented.

Our results indicate a decrease in the diversity of the gut microbiota in CD patients compared to healthy volunteers, which has also been found in many other studies [37,38,39]. Changes in the abundance of the families *Bacteroidaceae*, *Prevotellaceae*, *Clostridiaceae*, *Lachnospiraceae*, *Ruminococcaceae*, *Eryspelotrichaceae*, *Enterobacteriaceae*, *Fusobacteriaceae*, *Lactobacillaceae*, *Enterococcaceae*, and *Streptococcaceae* are often detected. The families *Bacteroidaceae*, *Prevotellaceae*, and *Rikenellaceae* are members of the phylum *Bacteroidetes* and perform several important functions in the gut, including metabolizing proteins and carbohydrates [40], producing butyrate [41], and preventing the colonization of the gastrointestinal tract by pathogenic bacteria [42]. In our study, among the most abundant phylum *Bacteroidetes* in CD patients, only the *Rikenellaceae* family decreased significantly compared to controls. The functions of this family in the gut microbiota have not yet been studied, but there is evidence of its decrease in patients with IBD and an increase in patients with irritable bowel syndrome [43]. Among the representatives of the phylum *Firmicutes*, there was a decrease in the proportion of the order *Clostridiales* and, in particular, of the family *Clostridiaceae*. They are known to be SCFAs producers and are involved in the metabolism of bile acids. There is a number of conflicting data on this taxon. While some authors observe an increase of *Clostidiaceae* in healthy controls and a decrease in CD patients [44,45,46,47], the others found an increase of this taxon in IBD patients [48,49]. In our study, we found a decreased abundance of the *Coriobacteriaceae* family of the *Actinobacteria* phylum in CD patients, which is consistent with previous studies [50,51,52]. *Coriobacteriaceae* have important functions in the gut including the conversion of bile salts and steroids and the activation of dietary polyphenols [53]. 

An increase of the *Enterobacteriaceae* family members was also found in the microbiota of CD patients. This is consistent with previously reported data in which the increased representation of this family was a marker of dysbiosis in IBD [8,54]. However, no association of any *E. coli* virulence genes with CD was found in the Russian population [55]. In our study, we found an increase in the proportion of lactic acid producing bacteria from the *Lactobacillaceae*, *Enterococcaceae*, and *Streptococcaceae* families in patients with CD, which is consistent with previous studies [9,56,57,58,59,60]. These bacteria are commensals; however, they can sometimes cause inflammation of various tissues in the respiratory, cardiovascular, and nervous systems [61,62,63,64]. Streptococci are known to provoke intestinal inflammation by inducing a pro-inflammatory response to lipoproteins and other components, as well as to the interaction of subtilisin-like protease (SspA) with Toll-like receptor 2 (TLR2) [65]. The role of enterococci in the pathogenesis of IBD has been described in a study showing that Enterococcus faecalis can cause IBD in the IL-10 knockout mouse model [66]. A pathogenicity island encoding surface aggregating protein (asa1), gelatinase (gelE), cytolysin (cylA), extracellular surface protein (esp), and hyaluronidase (hyl) was also identified as a possible trigger of the host inflammatory response [67]. Whether lactobacilli can provoke IBD or are simply adapted to survive in an inflamed gut is still an open question.

Many studies attempted to identify bacterial taxa that change with IBD activity/severity. Many of them agree that *Faecalibacterium prausnitzii* is associated with minimal inflammation [68,69,70]. However, the results for other taxa are conflicting. We found an increase in the representation of the *Enterococcaceae* and *Micrococcaceae* families in the gut microbiota of patients with more severe CD, which is consistent with the results of other studies [71,72,73]. In addition, we found a decrease in the abundance of the *Eryspelotrichaceae* and *Coriobacteriaceae* families with higher disease activity. A similar trend was observed for the *Eryspelotrichaceae*, while the opposite was found for *Coriobacteriaceae* by Papa et al. [74]. In our study, similar to Tedjo et al. *Bacteroidaceae* were increased in patients with mild CD [75], whereas other authors found the opposite [46,75,76]. Therefore, there is no clear understanding of the microbiota composition of IBD patients according to disease severity.

According to our data, the microbiota of CD patients is heterogeneous and two types of communities that can be identified. Thus, patients with a type I microbiota community shared it with control samples. Patients with a type II microbiota community are characterized by a lower diversity of the microbiota and a lower number of observed OTUs compared with the type I, indicating a more severe dysbiosis. In a study by Vieira-Silva et al., a similar method revealed four enterotypes, whose drivers were *Ruminococcaceae*, *Prevotella*, and *Bacteroides* [70]. The microbiota enterotypes of the Japanese, European, and American populations are characterized by the same taxa [77]. Other enterotypes were identified in a model organism study by Barron et al. where the main driver taxa were *Lachnospiraceae* and *Ruminoccoacceae*, *Enterobacteriaceae* and *Lactobacillus*, *Erysipelotrichaceae* and *Akkermansia* [78]. In our study, a number of bacterial families were represented differently in the microbiota community types. Thus, study participants with community type II had an increased abundance of *Enterobacteriaceae*, *Enterococcaceae*, and *Streptococcaceae* families, which, as noted above, are typical characteristics of CD patients’ gut microbiota. In addition, the abundance of *Verrucomicrobiaceae*, whose role in the pathogenesis of IBD is actively debated, was increased. Some authors noted a decrease of its representation in IBD and even suggested the use of *Akkermansia muciniphila* as a new generation probiotics [9,78,79,80], while others showed its increase in the microbiota of CD patients and suggested that it degrades the mucin of the intestinal mucosa, thereby provoking its inflammation [8]. There was also a decrease in the abundance of *Bacteroidaceae*, *Prevotellaceae*, *Lachnospiraceae*, *Ruminococcaceae*, *Erysipelotrichaceae*, and unclassified *Clostridiales* in the CD patients’ microbiota community of type II. These bacteria belong to the normal microbiota and are important in keeping the gut healthy. Thus, the second type of microbial community is characterized with more prominent dysbiotic changes in the microbiota of CD patients.

We found no differences in the clinical characteristics of CD (duration of disease, location of inflammation, disease activity, and stool frequency) between the two groups of patients with different types of gut microbiota communities, suggesting the presence of other reasons for this distribution.

As CD is a multifactorial disease, genetic factors may be responsible for differences in the gut microbiota composition. There are 24 single nucleotide polymorphisms studied, which have previously been associated with CD in various populations. Compared with controls, patients with CD have a significantly higher allele frequency of 8 SNPs. For the remaining 16 SNPs, no significant differences were found, probably due to the regional characteristics of the Russian population o the limited size of the cohort. It is known that the representation of some bacterial taxa in the intestinal microbiota is associated with specific alleles of host SNP. Therefore, polymorphisms in the *LCT* gene determine the percentage of *Bifidobacterium* in the gut microbiota of healthy individuals [81], which can be explained by bacterial enzymes compensating for lactase deficiency. There are also data on the relationship of representatives of *Akkermansia*, *Anaerostipes*, *Clostridiaceae*, *Blautia*, *Dialister*, *Bacteroides*, *Atopobium*, etc. with various host genetic loci, but the mechanism of these relationships has not been studied [81,82,83]. In the case of IBD, a high abundance of *Enterobacteriaceae* was found in the microbiota of NOD2-deficient patients [84]. Certain polymorphisms in the *FUT2* gene were associated with decreased SCFAs-producing *Faecalibacterium* and increased *Proteobacteria* [85]. It is also known that the *ATG16L1* T300A variant is associated with increased abundance of the *Bacteroides* genus [17]. In our study, rs9858542A allele in the *BSN* gene was found to be more frequent in CD patients with a second dysbiotic type of microbiota community and negatively correlated with the number of observed OTUs and *Bacteroidetes* phylum representation. The rs9858542A allele is known to be associated with an increased risk of CD [27,28,29]. The *BSN* gene encodes Bassoon Presynaptic Cytomatrix Protein, which is involved in organizing the presynaptic cytoskeleton and expressed primarily in brain neurons, although there is an evidence that this protein is also expressed at low levels in enteroendocrine cells in the gastrointestinal tract, including the stomach, duodenum, colon, and rectum [86]. These cells produce gut hormones that control digesting and food absorbtion, insulin secretion, etc. [87]. It is also known that the gut microbiota produce several metabolites (SCFAs, secondary bile acids, indoles, and lipopolysaccharides) that stimulate enteroendocrine cells [88,89,90,91,92]. The mechanism of *BSN* gene product interaction with intestinal microbiota is still unknown, but probably involves the interplay of microbiota metabolites with host enteroendocrine cells. In our study, we also found that the T allele in rs3816769 of the *STAT3* gene is significantly more frequent in CD patients with a second dysbiotic type of intestinal microbiota and negatively correlates with the *Bacteroidetes* phylum and *Bacteroidaceae* family in particular. This variant is also known to be associated with CD risk [30,93]. The transcription factor STAT3 (signal transducer and activator of transcription 3) regulates apoptosis, cell growth and inflammation in response to internal and external stimuli. In animal models, STAT3 activation in intestinal epithelial cells is required for wound healing, but also leads to the development of colitis-associated cancer in chronic inflammation [92,94]. Additionally, STAT3-deficient mice have increased sensitivity to bacterial lipopolysaccharide and increased levels of pro-inflammatory cytokines, and are more prone to chronic enterocolitis [95]. Zhao et al. found that microbial SCFAs activate STAT3 in intestinal epithelial cells, while STAT3 knockout resulted in a decrease in SCFA-induced antimicrobial peptide production [96]. Therefore, the *STAT3* gene mutation rs3816769T may affect the host-microbiota interaction. The C allele of rs1793004 in the *NELL1* gene was significantly more frequent in CD patients with the second dysbiotic microbiota type. Furthermore, a negative correlation of this variant with the *Ruminococcaceae* family and a positive correlation with *Enterococcaceae* were found. *NELL1* encodes neural epidermal growth factor-like 1, which is expressed at significant levels in epithelial cells of the small and large intestine, including inflamed epithelium [97]. The association of rs1793004C with IBD has been demonstrated by a genome-wide association study in a German population of IBD patients [97]. However, the mechanisms by which the *NELL1* gene product interacts with the intestinal microbiota remain unknown.

The findings of this study regarding the association between genetic polymorphisms and intestinal microbiota composition may help in developing personalized therapy for CD patients. Probiotics are considered a promising treatment of various autoimmune diseases–type 1 diabetes [98], multiple sclerosis [99], autoimmune hepatitis [100], rheumatoid arthritis [101], etc. Such therapy may include traditional probiotics (based on lactobacilli and bifidobacteria), next generation probiotics (based on *Faecalibacterium prausnitzii* [102] or *Akkermansia muciniphila* [103]), or fecal microbiota transplantation [104].

The limitation of the study is the relatively small number of healthy volunteers. However, differences in the microbiota of CD patients and healthy controls have been described in many previous works, while the variability of the microbiota within a group of CD patients is much less discussed. For this reason, we decided to study a larger number of CD patients for more reliable results. Taking these limitations into account, further investigations on associations of microbiota and genetic markers in both CD patients and healthy controls are required.

## 4. Materials and Methods

### 4.1. CD Patient and Controls

Venous blood and stool samples were collected from CD patients admitted to the Republican Clinical Hospital (Kazan, Russia) and clinical department of the Lopukhin Federal Research and Clinical Center of Physical-Chemical Medicine of Federal Medical Biological Agency (Moscow, Russia) during the period 2017–2021 (43/53 male/female, 32.3 ± 11.8 years old). CD was diagnosed using standard clinical, endoscopic, and histological criteria. The control group consisted of 24 volunteers (9/15 male/female, 35.3 ± 10.0 years old) from the same regions of Russia as the CD patients. Eligibility of patients with CD and healthy volunteers was determined according to specific inclusion/exclusion criteria as listed in Appendix A.

### 4.2. Ethics Statement 

Informed consent was obtained from all subjects involved in the study. The study was conducted in accordance with the recommendations of the local ethics committee of the Kazan Federal University, Kazan, Russia (Protocol No. 6, dated 13 October 2017) and Interuniversity ethics committee, Moscow, Russia (Protocol No.8, dated 23 September 2021).

### 4.3. 16S rRNA Gene-Based Metagenomic Analysis of Stool Samples

Genomic DNA was extracted from fecal samples using the QIAamp DNA Stool Mini Kit (Qiagen, Germantown, MD, USA) in accordance with the manufacturer’s instructions. A 16S rRNA sequencing library was constructed according to the 16S metagenomics sequencing library preparation protocol (Illumina, San Diego, CA, USA) targeting the V3 and V4 hypervariable regions of the 16S rRNA gene. The initial PCR was performed with template DNA using region-specific primers shown to have compatibility with the Illumina index and sequencing adapters (forward primer: 5′-TCGTCGGCAGCGTCAGATGTGTATAAGAGACAGTCGTCGGCAGCGTCAGATGTGTATAAGAGACAGCCTACGGGNGGCWGCAG-3′; reverse primer: 5′-GTCTCGTGGGCTCGGAGATGTGTATAAGAGACAGGTCTCGTGGGCTCGGAGATGTGTATAAGAGACAGGACTACHVGGGTATCTAATCC-3′). After purification of PCR products with AMPure XP magnetic beads, the second PCR was performed using primers from a Nextera XT Index Kit (Illumina). Subsequently, purified PCR products were visualized using gel electrophoresis and quantified with a Qubit dsDNA HS Assay Kit (Thermo Scientific, Waltham, MA, USA) on a Qubit 2.0 fluorometer. The sample pool (4 nM) was denatured with 0.2 N NaOH, diluted further to 4 pM, and combined with 20% (*v*/*v*) denatured 4 pM PhiX, prepared following Illumina guidelines. Sequencing of 16S rRNA gene V3-V4 variable regions was performed on the Illumina MiSeq platform in 2 × 300 bp mode at the Interdisciplinary Center of Shared Use of Kazan Federal University. 

Reads were further processed and analyzed using the QIIME software, version 1.9.1 [105] according to protocols. Before filtering, there were 53,175–182,362 (median 91,061) read pairs per sample. Paired-end reads were initially merged and then processed to remove low quality and chimeric sequence data. The rarefaction step was performed to reduce sequencing depth heterogeneity between samples. After quality filtering, chimera filtering and rarefying, we analyzed on average 20,938 joined read pairs. Sequences were clustered into operational taxonomic units (OTU) based on the 97% identity threshold (open reference-based OTU picking strategy); the SILVA database v.138 [106] was used. To characterize the richness and evenness of the bacterial community, the alpha diversity index was calculated using Shannon’s metrics.

### 4.4. Genotyping 

A total of 24 SNPs were selected based on data indicating their potential association with risk for IBD (Appendix A). Genomic DNA from venous blood was isolated and purified using the QIAamp DNA Mini Kit (Qiagen, Germantown, MD, USA) as described by the manufacturer. PCR amplification was performed using the primers listed in Appendix A according to the protocol [107]. Genotyping was performed using MALDI-TOF mass spectrometry as described previously [107].

### 4.5. Statistical Analysis 

The distribution of genotypes for all SNPs was tested for compliance with the Hardy-Weinberg equilibrium using the chi-square test. Analysis of the allele frequencies was done using Fisher’s exact test. The strength of associations was assessed using the odds ratio (OR, (lower 95% confidence interval; upper 95% confidence interval)). Differences in the taxonomic composition of the gut microbiota were assessed using the Kruskal-Wallis test. Correlations between genotypes and gut microbiota composition were analyzed using the R “psych” package [108] based on the Spearman’s rank correlation coefficient using an additive genetic model (depending on the genotype, a higher risk of developing CD corresponds to a higher rank). *p* < 0.05 values were considered as significant. To determine the types of bacterial communities, the Dirichlet multinomial mixture algorithm was applied to cluster the gut microbiota samples [109].

## Figures and Tables

**Figure 1 ijms-24-07998-f001:**
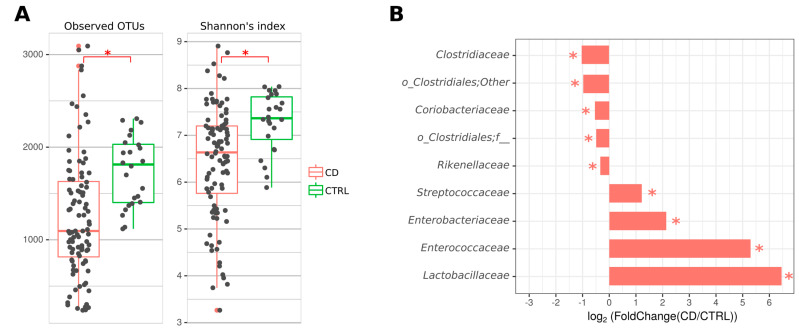
Analysis of gut microbiota taxonomic composition of CD patients and healthy volunteers. (**A**)—Number of OTUs and Shannon’s diversity index per group. (**B**)—Most abundant bacterial families significantly differentiated between comparison groups. *—*p* < 0.05 (Kruskall-Wallis test).

**Figure 2 ijms-24-07998-f002:**
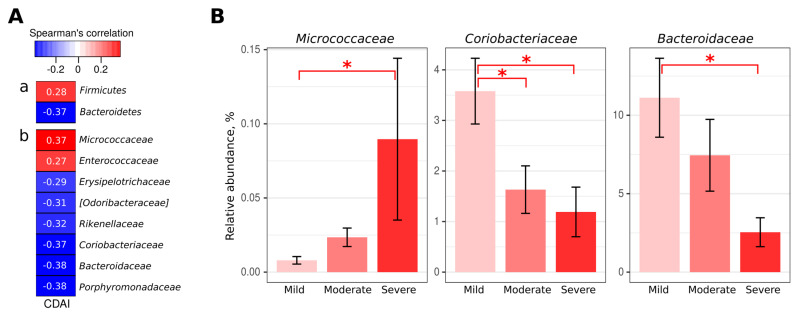
Analysis of gut microbiota taxonomic composition of CD patients with different disease activity. (**A**)—Statistically significant Spearman’s correlations between CDAI and gut microbiota taxonomic composition (*p* < 0.05). a—Bacterial phyla, b—Bacterial families. (**B**)—Most abundant bacterial families significantly differentiated between comparison groups. *—*p* < 0.05 (Kruskall-Wallis test with Benjamini-Hochberg correction for multiple comparisons).

**Figure 3 ijms-24-07998-f003:**
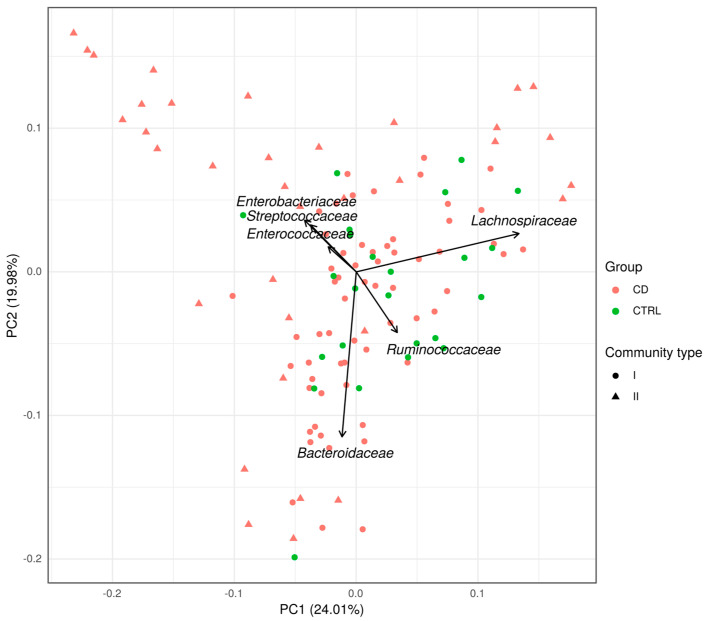
Principal component analysis based on bacterial composition of gut microbiota on family level.

**Figure 4 ijms-24-07998-f004:**
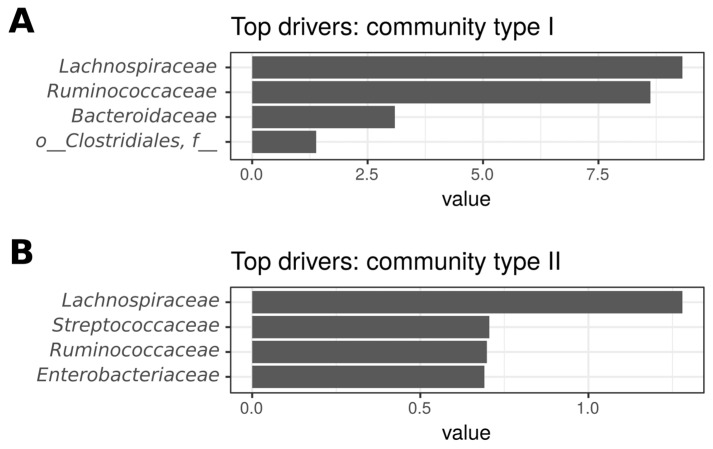
Top four driver families in different community types of gut microbiota. (**A**)—I community type; (**B**)—II community type.

**Figure 5 ijms-24-07998-f005:**
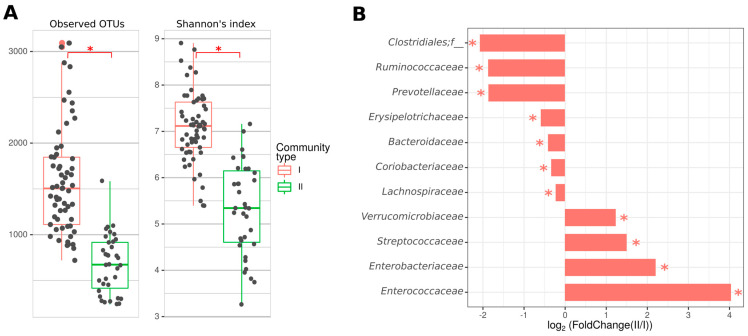
Analysis of gut microbiota taxonomic composition of CD patients with different types of gut microbiota community. (**A**)—Number of OTU and Shannon’s diversity index per group. (**B**)—Most abundant bacterial families statistically significantly differentiated between comparison groups. *—*p* < 0.05 (Kruskall-Wallis test).

**Figure 6 ijms-24-07998-f006:**
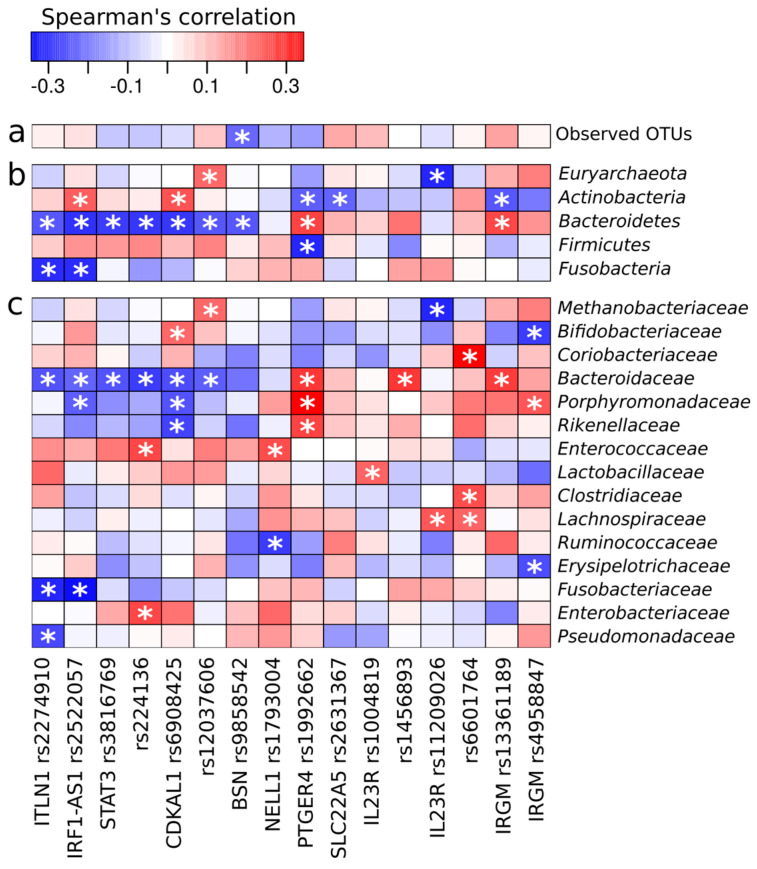
Spearman’s correlations between genetic polymorphisms and gut microbiota taxonomic composition of CD patients. (**a**)—Number of OTU, (**b**)—Bacterial phyla, (**c**)—Bacterial families. *—*p* < 0.05.

**Table 1 ijms-24-07998-t001:** Clinical characteristics of CD patients.

Clinical Characteristics	% of Samples (Total *n* = 96)
Location of inflammation	Ileitis—13.5% Colitis—47.9% Ileocolitis—38.6%
Phenotypic subtype	Inflammatory—33.3% Stricturing—55.2% Fistulizing—11.5%
Crohn’s disease activity index (CDAI)	Mildly active (150–220 points)—68.75% Moderately active (221–450 points)—25% Severely active (>451 points)—6.25%
Therapy: no treatments (0) 5-aminosalicylic acid (1) steroids (2) immunosuppressor (3) biologics (4)	(0)—7.29% (1)—28.13% (2)—5.21% (3)—9.38% (1) + (2)—5.21% (1) + (3)—2.08% (1) + (4)—4.17% (1) + (2) + (3)—1.04% (1) + (2) + (4)—2.08% (1) + (3) + (4)—2.08% (2) + (3)—6.25% (2) + (4)—8.33% (2) + (3) + (4)—4.17% (3) + (4)—14.58%

**Table 2 ijms-24-07998-t002:** Clinical characteristics of CD patients with different types of communities.

Clinical Characteristics	Mean ± SD		* p * Value (*-Kruskal-Wallis Test, ^§^-Exact Fisher Test)
	Community Type I	Community Type II	
CD duration, years	9.0 ± 4.4	9.7 ± 4.7	0.75 *
Number of stools per day, *n*	1.5 ± 1.2	1.8 ± 1.3	0.45 *
Crohn’s disease activity index (CDAI)	245.0 ± 77.2	274.8 ± 204.2	0.86 *
Body Mass Index	21.2 ± 4.1	24.2 ± 7.0	0.22 *
Location of inflammation (ileitis/colitis/ileocolitis), %	9.7/58.1/32.3	17.4/34.8/47.8	0.27 ^§^
Phenotypic subtype (inflammatory/stricturing/fistulizing), %	32.3/54.8/12.9	34.8/56.5/8.7	1.00 ^§^

**Table 3 ijms-24-07998-t003:** Allelic distribution of 24 SNPs in CD patients and healthy volunteers.

SNP (Gene)	Alleles	CD Patients (% of Alleles)	Healthy Volunteers (% of Alleles)	OR (Lower 95% CI; Upper 95% CI)	*p* Value, Exact Fisher Test
rs2241880 (*ATG16L1*)	A/G	67.1/32.9	47.8/52.2	0.45 (0.23; 0.88)	** 0.024 **
rs9858542 (*BSN*)	A/G	33.9/66.1	29.2/70.8	0.81 (0.39; 1.59)	0.609
rs6908425 (*CDKAL1*)	C/T	75.5/24.5	68.8/31.3	0.71 (0.36; 1.46)	0.359
rs6596075 (*IBD5*)	C/G	89.6/10.4	83.3/16.7	0.58 (0.24; 1.50)	0.219
rs11805303 (*IL23R*)	C/T	69.3/30.7	58.3/41.7	0.62 (0.32; 1.21)	0.171
rs1004819 (*IL23R*)	A/G	47.9/52.1	29.2/70.8	0.45 (0.22; 0.88)	** 0.023 **
rs10489629 (*IL23R*)	C/T	36.5/63.5	47.9/52.1	1.60 (0.84; 3.05)	0.185
rs11209026 (*IL23R*)	A/G	9.4/90.6	27.1/72.9	3.57 (1.57; 7.99)	** 0.003 **
rs2522057 (*IRF1-AS1*)	C/G	81.8/18.2	70.8/29.2	0.54 (0.26; 1.14)	0.109
rs13361189 (*IRGM*)	C/T	10.9/89.1	6.3/93.8	0.57 (0.12; 1.76)	0.428
rs4958847 (*IRGM*)	A/G	32.3/67.7	14.6/85.4	0.37 (0.14; 0.82)	** 0.020 **
rs2274910 (*ITLN1*)	C/T	66.7/33.3	37.5/62.5	0.30 (0.15; 0.58)	** <0.001 **
rs1793004 (*NELL1*)	C/G	18.8/81.3	29.2/70.8	1.79 (0.85; 3.64)	0.116
rs2836878 (*PSMG1*)	A/G	6.8/93.2	2.1/97.9	0.32 (0.01; 1.67)	0.313
rs1992662 (*PTGER4*)	A/G	33.3/66.7	56.3/43.8	2.56 (1.34; 4.93)	** 0.005 **
rs8111071 (*RSPH6A*)	A/G	91.1/8.9	95.8/4.2	2.09 (0.57; 14.73)	0.380
rs2631367 (*SLC22A5*)	C/G	55.7/44.3	39.6/60.4	0.52 (0.27; 0.99)	0.053
rs3816769 (*STAT3*)	C/T	22.9/77.1	22.9/77.1	1.01 (0.45; 2.10)	1.000
rs7753394 (*TNFAIP3*)	C/T	53.1/46.9	39.6/60.4	0.58 (0.30; 1.10)	0.108
rs1456893 (intergenic)	A/G	65.6/34.4	72.9/27.1	1.40 (0.70; 2.93)	0.393
rs224136 (intergenic)	C/T	86.0/14.0	87.5/12.5	1.07 (0.27; 7.90)	1.000
rs6601764 (intergenic)	C/T	41.1/58.9	64.6/35.4	2.59 (1.35; 5.11)	** 0.006 **
rs7807268 (intergenic)	C/G	37.0/63.0	6.3/93.8	0.12 (0.03; 0.34)	** <0.001 **
rs12037606 (intergenic)	A/G	42.2/57.8	31.3/68.8	0.63 (0.31; 1.22)	0.190

**Table 4 ijms-24-07998-t004:** Allelic distribution of 3 SNPs with significantly differentiated occurrence in CD patients with different types of gut microbiota communities.

SNP (Gene)	Alleles	Community Type II (% of Alleles)	Community Type I (% of Alleles)	OR (Lower 95% CI; Upper 95% CI)	*p* Value, Exact Fisher Test
rs9858542 (*BSN*)	A/G	45.7/54.3	27.0/73.0	0.44 (0.24;0.82)	**0.011**
rs3816769 (*STAT3*)	C/T	14.3/85.7	27.9/72.1	2.29 (1.08; 5.24)	**0.033**
rs1793004 (*NELL1*)	C/G	27.1/72.9	13.9/86.1	0.44 (0.21;0.92)	**0.034**

## Data Availability

The data presented in this study are available on request from the corresponding author. Raw reads are deposited in the NCBI SRA under accession number PRJNA938107 in the fastq format (https://www.ncbi.nlm.nih.gov/sra/PRJNA938107, accessed on 30 March 2023).

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
