# Peer review of "Association between Taxonomic Composition of Gut Microbiota and Host Single Nucleotide Polymorphisms in Crohn’s Disease Patients from Russia"

_ijms, 2023, doi:10.3390/ijms24097998_

Round 1

Reviewer 1 Report

In this study, the Authors tried to assess the taxonomic composition of the gut microbiota and genotypes of 24 single nucleotide polymorphisms (SNP) associated with the risk of Crohn's disease (CD). They studied 96 CD patients and 24 healthy volunteers from Russia. Statistically significant differences were found in the allele frequencies for 8 SNPs and in the taxonomic composition of the gut microbiota in CD patients compared to controls. Of interest, two types of gut microbiota communities were identified in CD patients. The main distinguishing driver bacterial families for the first community type were Bacteroidaceae and unclassified members of the Clostridiales order, and the second type was characterized by increased abundance of Streptococcaceae and Enterobacteriaceae. Differences in the allele frequencies of the rs9858542 (BSN), rs3816769 (STAT3) and rs1793004 (NELL1) were also found between groups of CD patients with different types of microbiota community. The authors concluded that their findings confirm the multifactorial nature of CD.

The study is of interest providing novel findings of potential pathogenic interest. The authors should further discuss the role of immune response to gut microbiota. In particular, it is well-known that CD is characterized by the appearance of antibodies to Anti-Saccharomyces cerevisiae (ASCA). The interaction between the immune system and gut microbiota has been reported also in other chronic intestinal disorders such as celiac disease where the positivity for ASCA has been found in a significant rate of patients at the diagnosis but also in patients with silent and asymptomatic celiac disease, likely as a result of increased intestinal permeability, as previously demonstrated (Anti-Saccharomyces cerevisiae and perinuclear anti-neutrophil cytoplasmic antibodies in coeliac disease before and after gluten-free diet. Aliment Pharmacol Ther. 2005 Apr 1;21(7):881-7;  Anti-saccharomyces cerevisiae antibodies (ASCA) in coeliac disease. Gut. 2006 Feb;55(2):296). The authors should discuss the increased intestinal permeability hypothesis as suggested by the development of immune response against saccharomyces cerevisiae.

-I would also suggest to tried to discuss the potential therapeutic impact of the study findings such as therapeutic interventions able to modify the microbiota as suggested also for autoimmune intestinal and extraintestinal diseases, as suggested (Editorial: gut microbiota profile in patients with autoimmune hepatitis-a clue for adjunctive probiotic therapy? Aliment Pharmacol Ther. 2020 Jul;52(2):392-394.).

Minor editing of English language required

Author Response

In this study, the Authors tried to assess the taxonomic composition of the gut microbiota and genotypes of 24 single nucleotide polymorphisms (SNP) associated with the risk of Crohn's disease (CD). They studied 96 CD patients and 24 healthy volunteers from Russia. Statistically significant differences were found in the allele frequencies for 8 SNPs and in the taxonomic composition of the gut microbiota in CD patients compared to controls. Of interest, two types of gut microbiota communities were identified in CD patients. The main distinguishing driver bacterial families for the first community type were Bacteroidaceae and unclassified members of the Clostridiales order, and the second type was characterized by increased abundance of Streptococcaceae and Enterobacteriaceae. Differences in the allele frequencies of the rs9858542 (BSN), rs3816769 (STAT3) and rs1793004 (NELL1) were also found between groups of CD patients with different types of microbiota community. The authors concluded that their findings confirm the multifactorial nature of CD.

Point 1: The study is of interest providing novel findings of potential pathogenic interest. The authors should further discuss the role of immune response to gut microbiota. In particular, it is well-known that CD is characterized by the appearance of antibodies to Anti-Saccharomyces cerevisiae (ASCA). The interaction between the immune system and gut microbiota has been reported also in other chronic intestinal disorders such as celiac disease where the positivity for ASCA has been found in a significant rate of patients at the diagnosis but also in patients with silent and asymptomatic celiac disease, likely as a result of increased intestinal permeability, as previously demonstrated (Anti-Saccharomyces cerevisiae and perinuclear anti-neutrophil cytoplasmic antibodies in coeliac disease before and after gluten-free diet. Aliment Pharmacol Ther. 2005 Apr 1;21(7):881-7;  Anti-saccharomyces cerevisiae antibodies (ASCA) in coeliac disease. Gut. 2006 Feb;55(2):296). The authors should discuss the increased intestinal permeability hypothesis as suggested by the development of immune response against saccharomyces cerevisiae.

Response 1: Thanks for your valuable comment. We have added a discussion of intestinal permeability to the Introduction chapter on the lines 59-70.

Point 2:-I would also suggest to tried to discuss the potential therapeutic impact of the study findings such as therapeutic interventions able to modify the microbiota as suggested also for autoimmune intestinal and extraintestinal diseases, as suggested (Editorial: gut microbiota profile in patients with autoimmune hepatitis-a clue for adjunctive probiotic therapy? Aliment Pharmacol Ther. 2020 Jul;52(2):392-394.).

Response 2: In the Discussion chapter on the lines 338-345, we have added points about probiotics for autoimmune diseases.

We are grateful for your evaluation of our study, we appreciate the time that you have spent on the review.

Reviewer 2 Report

This is clearly written and well-organized manuscript. Despite of the small sample size, I consider that the manuscript might be relevant for the audience to read and to take into consideration for future research. I suggest to accept it in the present form.

Author Response

This is clearly written and well-organized manuscript. Despite of the small sample size, I consider that the manuscript might be relevant for the audience to read and to take into consideration for future research. I suggest to accept it in the present form.
Response: We are grateful for your evaluation of our study, we appreciate the time that you have spent on the review.

Reviewer 3 Report

The manuscript presents the results on the role of Gut microbiota and SNPs in patients with Crohn’s disease. This is an interesting study, however, there are several major comments:

Authors should provide information on whether patients involved in the study received any treatment. And if so, authors should add this information. Authors should clearly state inclusion and exclusion criteria. 

In the methodology section it says that venous blood was used and CD was diagnosed using clinical testing. Did authors measured any association between expression of immune cells and gut microbiota and SNPs in CD patients?

Why authors did not compare the gut microbiota with different groups of CD Patients ( mild, moderate, severe)?

How authors corrected the differences between high number of patients (n=96) and number of healthy volunteers (n=24)?

Author Response

The manuscript presents the results on the role of Gut microbiota and SNPs in patients with Crohn’s disease. This is an interesting study, however, there are several major comments:

 Point 1: Authors should provide information on whether patients involved in the study received any treatment. And if so, authors should add this information. Authors should clearly state inclusion and exclusion criteria. 

Response 1: Thank you for the comments. We have added information about the therapy to the Table 1. Inclusion/exclusion criteria for the study are listed in Supplementary Table 2.

Point 2: In the methodology section it says that venous blood was used and CD was diagnosed using clinical testing. Did authors measured any association between expression of immune cells and gut microbiota and SNPs in CD patients?

Response 2: Blood immune cell expression analysis is not required to diagnose Crohn's disease. CD was diagnosed based on the clinical and endoscopic findings and was proved by histology. Thus, we cannot analyze the association of immune cell expression with SNPs and gut microbiota because such data is not available for the majority of patients.

Point 3: Why authors did not compare the gut microbiota with different groups of CD Patients ( mild, moderate, severe)?

Response 3: We have added a comparison of the microbiota of CD patients according to disease activity in the Results (lines 103-117, Figure 2) and Discussion (lines 245-256) sections.

Point 4: How authors corrected the differences between high number of patients (n=96) and number of healthy volunteers (n=24)?

Response 4: No additional methods were used to correct unequal sample sizes between patients and controls. We used statistical tests that don't require the same number of samples in each comparison group. There is evidence that an unequal number of samples can cause false results, but only in comparisons between three or more groups [Brunner, Edgar, et al. "Ranks and Pseudo‐ranks—Surprising Results of Certain Rank Tests in Unbalanced Designs."International Statistical Review 89.2 (2021): 349-366]. In our study, we compared only 2 groups with unbalanced sample sizes - CD patients and controls. Also, we understand that statistical power is based on the smallest sample size. Having more samples in a larger group doesn't hurt statistical power, it just doesn't increase it.

We are grateful for your evaluation of our study, we appreciate the time that you have spent on the review.

Round 2

Reviewer 3 Report

The authors have addressed all the comments